# Obesity and Cancer Metastasis: Molecular and Translational Perspectives

**DOI:** 10.3390/cancers12123798

**Published:** 2020-12-16

**Authors:** Stephanie Annett, Gillian Moore, Tracy Robson

**Affiliations:** School of Pharmacy and Biomolecular Sciences, RCSI University of Medicine and Health Science, 123 St Stephen’s Green, Dublin D02 YN77, Ireland; stephanieannett@rcsi.com (S.A.); gillianmoore@rcsi.com (G.M.)

**Keywords:** obesity, adipose tissue, tumor progression, cancer relapse, metastasis, cytokines, adipokines, extracellular matrix, extra cellular vesicles, cancer metabolism

## Abstract

**Simple Summary:**

A major challenge in treating cancer is when the cancer spreads from its original site to other parts of the body, a process also known as metastasis. In order to survive, the cancer cells must communicate with their environment for survival. It is increasingly recognized that fat tissue supports the survival of metastatic cancer cells allowing tumors to form at distant sites. Obese patients therefore have a worse prognosis due to accelerated tumor spread. Preventing cancer cells from communicating with fat tissue could therefore lead to new treatments aimed at stopping this spread. In this review we discuss how dysfunctional fat tissue supports the metastatic process and evaluate new therapies and lifestyle interventions that aim to prevent the communication between cancer cells and fat tissue.

**Abstract:**

Obesity is a modern health problem that has reached pandemic proportions. It is an established risk factor for carcinogenesis, however, evidence for the contribution of adipose tissue to the metastatic behavior of tumors is also mounting. Over 90% of cancer mortality is attributed to metastasis and metastatic tumor cells must communicate with their microenvironment for survival. Many of the characteristics observed in obese adipose tissue strongly mirror the tumor microenvironment. Thus in the case of prostate, pancreatic and breast cancer and esophageal adenocarcinoma, which are all located in close anatomical proximity to an adipose tissue depot, the adjacent fat provides an ideal microenvironment to enhance tumor growth, progression and metastasis. Adipocytes provide adipokines, fatty acids and other soluble factors to tumor cells whilst immune cells infiltrate the tumor microenvironment. In addition, there are emerging studies on the role of the extracellular vesicles secreted from adipose tissue, and the extracellular matrix itself, as drivers of obesity-induced metastasis. In the present review, we discuss the major mechanisms responsible for the obesity–metastatic link. Furthermore, understanding these complex mechanisms will provide novel therapies to halt the tumor–adipose tissue crosstalk with the ultimate aim of inhibiting tumor progression and metastatic growth.

## 1. Introduction

The metabolic and cardiovascular risks of obesity are well known. However, it is estimated that 40% of all cancer deaths are also attributable to obesity [1]. Indeed, globally, excess body weight is third behind smoking and infection as an attributable risk factor for cancer, and second to smoking in Western populations [2]. Obesity adversely effects cancer in two ways, (i) by promoting carcinogenesis resulting in a higher cancer incidence and (ii) cancer progression resulting in an increased risk of mortality [3]. In breast cancer, for example, obesity is only associated with an increased incidence of post-menopausal breast cancer, whilst obesity is a risk factor for progression in all breast cancer subtypes [4]. The global obesity rate in women is projected to reach 21% by 2025 and this is particularly alarming considering that 55% of all female cancers have an obesity associated mechanism [3]. Central obesity, resulting from the overgrowth of visceral white adipose tissue (WAT), has been specifically linked to cancer progression [5]. While diet is undoubtedly important in obesity, animal models have indicated that WAT overgrowth directly promotes cancer progression irrespective of diet [3,6]. Epidemiological studies have demonstrated a compelling association between cancer risk and obesity. Analysis of the association between body mass index (BMI) and early stage breast cancer outcomes in the Danish Breast Cancer Cooperative Group (*n =* 53,816 women) revealed that obese women have a 46% higher risk of developing distinct metastasis at 10 year follow up compared to normal weight women [7]. Furthermore, a meta-analysis of 82 studies found a 41% and 35% higher risk, respectively, of all-cause mortality and breast cancer specific mortality in obese women compared to normal weight women [2]. An umbrella review of 204 meta-analyses revealed a strong association between obesity and gastrointestinal cancers including esophageal adenocarcinoma (OAC) [8]. OAC was notable for a progressive increase in risk ratio (RR) for each 5 kg/m^2^ increase in BMI (RR 4.8 for a BMI ≥ 40 kg/m^2^) suggesting a dose–response effect [9]. The top two cancers demonstrating very strong associations with BMI are endometrial (RR 1.48) and OAC (RR 1.45) [10]. 

In obesity, white adipocytes become hypertrophic and hyperplasic, which results in physiologic changes, including elevated free fatty acids (FFAs) and triglycerides, increased blood glucose and insulin resistance. Obese adipose tissue increases production of proinflammatory cytokines, e.g., tumor necrosis factor (TNFα), interleukin-6 (IL-6), interleukin-1β (IL-1β) and adipokines (e.g., leptin) [3]. Metastasis is the primary cause of cancer morbidity and mortality and efforts to unravel the molecular mechanisms linking dysfunctional adipose tissue and the ability of tumor cells to acquire metastatic properties will lead to the discovery of novel targets for metastasis. Here we review the recent findings regarding the molecular processes underpinning the impact of adipose tissue on cancer metastasis and potential therapeutic interventions.

## 2. The Metastatic Cascade 

The “hallmarks of cancer” define characteristics that are critical for cellular transformation [11]. Among the hallmarks there is only one defining factor, invasion, which distinguishes a malignant and benign tumor [11]. The metastatic cascade begins with local invasion before progressing to intravasation, arrest at distant organs, extravasation, micro metastasis formation and finally metastatic outgrowth (Figure 1).

In the early stages of tumor development, the primary tumor cell mass typically has an expansive phase in the absence of invasion, encapsulated in a dense fibrous network (i.e., desmoplasia) [12]. A subset of neoplastic cells acquire the ability to escape through the basement membrane and detach from the primary tumor [12]. The dissemination of cancer cells is a consequence of chromosomal instability that causes continuous errors in chromosome segregation during mitosis [13]. This in turns leads to the rupture of micronuclei and the secretion of genomic DNA into the cytosol, which activates DNA sensing pathways and NF-κB signaling [13]. In addition, epithelial–mesenchymal transition (EMT) is a transdifferentiating process, which permits epithelial cells to attain a mesenchymal phenotype with migratory potential [14]. Spontaneous EMT in primary tumor cells can be triggered by hypoxia, metabolic stressors and matrix stiffness [15,16]. Although some studies have cast doubts on the necessity of EMT during metastasis [17,18], there is evidence for cells expressing both epithelial and mesenchymal markers within the primary tumor, circulation and at a secondary metastatic site [19]. Therefore EMT is being increasingly understood as a spectrum of transitional stages between epithelial and mesenchymal phenotypes rather than a binary choice between an epithelial or mesenchymal phenotype [20,21]. There is evidence that cells in different EMT stages prefer certain microenvironments, for example, metastatic cells with a predominating mesenchymal phenotype proliferate near endothelial and inflammatory cells [21]. These tumor cells release factors to attract immune cells and stimulate angiogenesis, thus promoting an inflammatory and highly vascularized niche [21]. Although EMT is required for metastatic dissemination, the opposite process of mesenchymal to epithelial transition (MET) is required for metastatic colonization of distant sites [22]. There is also substantial evidence that disseminated tumor cells express stem cell markers, such as aldehyde dehydrogenase (ALDH), and functionally these cells, highly enriched in stem cells markers, have an enhanced ability to cause metastasis [19,23,24,25]. Furthermore, genome wide analysis of both cells undergoing EMT and circulating tumor cells has revealed a similar transcriptome to primary cancer stem cells (CSCs) thus indicating an overlapping subpopulation [19,26,27]. In pancreatic cancer, primary CD133^+^ CSCs demonstrated classic CSC characteristics such as tumor initiation and chemoresistance [28]. However, at the invasive front of the tumor CD133^+^ cells are enriched for CXC-chemokine receptor 4 (CXCR4) and the CD133^+^CXCR4^+^ population is more migratory than CD133^+^CXCR4^--^ [28]. Moreover, patients with increased CD133^+^CXCR4^+^ cells had more metastatic disease [28]. This indicates that microenvironmental cues within the tumor can trigger heterogeneity in CSCs and CD133^+^CXCR4^+^ and CD133^+^CXCR4^-^ are not a distinct subpopulations but a gradient of stemness phenotypes [19,28]. CSCs are more resistant to chemotherapy due to higher expression of multidrug resistance (MDR) or detoxification proteins such as aldehyde dehydrogenase (ALDH) [29].

Prior to exiting the primary tumor mass, tumor cells communicate with other microenvironments, termed the premetastatic niche, and this niche is selectively primed by secreted factors and extracellular vesicles to induce vascular leakage, extra cellular matrix (ECM) remodeling and immunosuppression [30]. Cancer cells also adjust the niche themselves by remodeling the ECM leading to stromal tumorigenesis [20]. Cancer patients release large numbers of cancer cells into the circulation daily, however animal studies of melanoma suggest that <0.1% of these cells metastasize [31]. The most widely studied route of dissemination is through the bloodstream (hematogenous), however metastatic cells may also migrate along nerves, lymphatic vessels, across coelomic cavities or along the basal side of endothelial cells and never enter the lumen [12]. Intravasation, the dissemination of cancer cells to organs through the lumen of the vasculature can be active or passive depending on the tumor type, microenvironment and vasculature [32]. The migration of metastatic cells into the circulation relies on chemokines and complement components that direct tumor cells through the vasculature and metabolic factors that result in an antioxidant effect [20]. Furthermore, the dissemination of circulating tumor cells (CTCs) is supported by a close association with immune cells such as activated platelets, macrophages and neutrophils [20]. When CTCs pass into small capillaries they become trapped leading to microvascular rupture or the cell undergoes extravasation [20]. Establishing a vascular network is required for metastatic colonization and this can occur though angiogenesis, co-opting existing vessels or inducing vasculogenic mimicry [20,33]. Furthermore, cancer cells can also exploit neuronal signaling pathways for growth and adaptation [20]. Cancer dormancy is an arrest phase that can occur after invasion into secondary sites and in some cancer survivors, dormant cells result in relapse long after the removal of the primary tumor [20,34]. The subsequent reactivation is governed by self-renewal pathways (e.g., Wnt, Hedgehog and Notch) and cells exhibit increased levels of stem cell associated genes [35]. While dormant cancer cells downregulate the expression of immune cell recognizable antigens, persistent host organ inflammation and the establishment of neutrophils in extracellular traps may transform dormant cells into aggressive metastasis [35,36]. In many of the steps from dissemination to colonization, the microenvironment plays a major role and this can be significantly altered by dysfunctional adipose tissue in people living with obesity.

## 3. Mechanisms Linking Adipose Tissue to the Metastatic Cascade

### 3.1. Adipocytes and Adipokines

Adipocytes secrete more than 600 soluble factors, known as adipokines, and the most well characterized are leptin and adiponectin [37]. Intra-abdominal cancers such as ovarian, colon and gastric, preferentially metastasize to the omentum, a peritoneal organ largely composed of adipocytes, suggesting that adipocytes significantly contribute to the metastatic cascade [38]. Omental adipocytes promote migration and invasion of ovarian cancer cells by secreting cytokines [39]. Neutralization of these cytokines reduced in vivo homing of ovarian cancer cells to mouse omentum, suggesting that adipocytes promote the early stages of metastasis [39]. In addition, ovarian cancer cells metabolically adapt to the increased availability of lipids by utilizing energy from fatty acids for growth [39]. Fatty acid binding protein 4 (FABP4) was strongly expressed at the adipocyte–cancer cell interface and pharmacological inactivation of FABP4 decreased cancer cell lipid accumulation, invasiveness and omental metastasis [39]. Circulating levels of FABP4 are markedly increased in obese individuals due to release from an expanded adipose tissue depot and FABP4 can induce mammary tumor stem cells by enhancing ALDH1 activity via IL-6/STAT3 signaling [40,41]. In addition, FABP4 promoted aggressive acute myeloid leukemia (AML) in obesity by enhancing aberrant DNA methyltransferase 1 (DNMT1)-dependent DNA methylation of tumor cells [42]. Upon interaction with cancer cells, adipocytes dedifferentiate into preadipocytes or are reprogrammed into cancer-associated adipocytes (CAAs), which resemble fibroblasts and have dispersed lipid droplets [43]. In breast cancer, matrix metalloproteinase (MMP)-11 is induced in adipocytes by adjacent invading cancer cells. In the presence of MMP-11, the activated adipocyte dedifferentiates into a preadipocyte fibroblast-like cell, which can sustain cancer cell invasion [44,45]. Bone marrow adipocytes constitute approximately 15% of bone marrow volume in young adults, rising to 60% by the age of 65 years old [46]. They have a distinctive phenotype, which resembles both white and brown adipose tissue, and they secrete fatty acids, cytokines and adipokines, which influence the whole bone microenvironment [47]. The bone provides a supportive microenvironment for both solid tumor and hematological metastasis, including breast, prostate and multiple myeloma and bone metastatic cancers primarily occur in older adults whose bone marrow is heavily populated by adipocytes [48,49]. Similarly to the omentum, cancer cells are attracted to the adipocytes in the metabolically active red marrow and this creates a niche in the bone marrow for disseminated cancer cells to establish and progress [48]. In addition, leukemic stem cells expressing the fatty acid transporter CD36, induce lipolysis in gonadal adipose tissue to support their metabolism and evade chemotherapy [50]. Lipids can also be trafficked between bone marrow adipocytes and cancer cells by upregulation of FABP4 and fuel growth and invasiveness in metastatic tumor cells [51]. In addition, bone marrow adipocytes have been shown to promote the Warburg phenotype in metastatic prostate cancer cells through oxygen independent HIF-1α activation [52]. Bone marrow adipocytes are a major source of circulating adiponectin, much greater than WAT [53]. Adiponectin is reported to suppress many elements of the early metastatic cascade including adhesion, invasion, migration and stem cell properties via numerous signaling pathways including WNT, NF-κB and JAK/STAT [54]. In advanced cancer associated with cachexia, hyperadiponectinemia has been observed [55]. In addition, increased adiponectin signaling in dendritic cells can blunt anti-tumor immune responses in patients with metastatic disease [56]. However, the late increase in adiponectin has very little influence on the course of the disease, as its role is thought to be more prominent in early metastatic spread [54]. During the development of obesity, preadipocytes differentiate incorrectly leading to hypoxia and the induction of hypoxia induced factor-1 (HIF-1) [57]. This inhibits the expression of adiponectin and increases the expression of leptin, resulting in a reduced adiponectin to leptin ratio in obesity-associated adipose tissue [58,59]. A high leptin to adiponectin ratio has been reported to increase the risk of postmenopausal and triple negative breast cancer (TNBC) progression [60,61]. Leptin is another adipokine important in tumor progression and secretion of leptin is increased in CAAs compared to mature adipocytes [43]. Leptin levels are increased in the plasma of post-menopausal breast cancer patients, which correlated with a higher grade, advanced tumor stages and presence of distant metastasis [37]. Leptin exerts it effect through the transmembrane leptin receptor and only the full length isoform, ObRb, contains the intracellular domain required for JAK/STAT signaling [62]. ObRb was significantly overexpressed in metastatic lymph nodes compared to the primary tumor in ER-breast cancer patients [63]. Furthermore, TNBC patient derived xenografts (PDX) grown in the presence of primary adipose stem cells (ASCs) isolated from obese donors (obASCs) had increased HLA1^+^ human tumor cells and CD44^+^CD24^−^ CSCs in the peripheral blood and metastasis compared to ASCs from lean women [64]. In addition, the knockdown of leptin expression in obASCs suppressed the prometastatic effect [64]. There is a plethora of evidence that leptin induces cell migration and invasion in breast cancer via JAK/STAT3 signaling [62]. In addition, the Notch signaling pathway is a key regulator of leptin induced cell migration in breast cancer obesity models [63]. Leptin has also been shown to promote the CSC phenotype though STAT3 activation and this in turn recruited a histone methyl transferase causing a repression of miR-200c by epigenetic silencing and the expansion of CSCs [65]. Autotaxin (ATX) is an adipocyte-derived lysophospholipase D that catalyzes the hydrolysis of circulating or cell associated lysophosphatidylcholine (LPC) to the bioactive lipid lysophosphatidic acid (LPA) [66]. Some tumor cells, such as in melanoma, glioblastoma and thyroid cancer, directly secrete ATX leading to a chronic inflammatory state and decreased acquired immune response [67]. In contrast, other tumor cells, such as breast, ovarian and pancreatic do not produce ATX and instead the tumor microenvironment is the primary source [67]. Interestingly, the ATX gene *(ENPP2)* was the second most upregulated gene in breast CSCs treated with paclitaxel whilst the LPP2 gene *(PLPP2)* was downregulated, indicating CSCs may favor an LPA-enriched microenvironment [68]. In a tissue microarray (TMA) of metastatic breast cancer, stromal ATX was highly expressed in bone metastasis [69]. In mice approximately 40% of ATX is produced by adipocytes and this increases when mice are fed a high fat diet [70]. Similarly, in humans ATX production increases in obesity, particularly in inflamed adipose tissue, and contributes to comorbidities including insulin resistance, hepatic steatosis and atherosclerosis [66,71,72,73]. In breast cancer cells both ATX and LPA are associated with mobility and invasive capacity via the JAK/STAT3 pathway or PI3K/MAPK pathways [74,75]. Therefore cross talk between adipose tissue derived ATX and tumors cells is a potential mechanism for tumor progression [76,77]. Indeed preclinical studies of an ATX inhibitor, ONO-843050 suppresses tumor growth and 60% of lung metastasis in a breast cancer mouse model [78]. Targeting other molecules in the ATX-LPA signaling pathways also results in decreased breast cancer metastasis formation in murine models [77,79,80]. The physiological upregulation of ATX occurs in response to inflammation and chronic activation of ATX-LPA signaling occurs in diseases such as pulmonary fibrosis, rheumatoid arthritis and inflammatory bowel diseases [81]. A first in class ATX inhibitor, GLPG1690, attenuated idiopathic pulmonary fibrosis in a Phase IIa clinical trial (NCT02738801) and two Phase III clinical trials are currently underway for this indication (NCT03711162; NCT03733444) [82,83]. In a preclinical breast cancer model, GLPG1690 acted synergistically with chemotherapy and radiotherapy to improve outcomes [84]. Considering the role of ATX in regulating CSCs and mobility, targeting ATX with inhibitors such as GLPG1690 may also prove to be beneficial in targeting metastasis, particularly in patients living with obesity, by preventing adipose tissue cross talk.

### 3.2. Immune Cells and Inflammatory Factors

CAAs secrete more inflammatory factors, such as monocyte chemoattractant protein (MCP-1), RANTES, IL-1β, IL-6 and TNFα than “normal” adipocytes and this can promote invasion and metastasis formation [43,85]. Furthermore, the recruitment and activation of immune cell subsets, particularly M1 macrophages, in obese WAT increases local and systemic levels of proinflammatory cytokines [86]. MCP-1 is elevated in obesity and secreted by many cells including tumor cells, fibroblasts, tumor infiltrating monocytes and endothelial cells [43,87]. In cancer cells, MCP-1 induces the expression of NOTCH1 and subsequently promotes the activity of CSCs and neovascularization [87,88]. RANTES expression in the peritumoral adipose tissue of women with TNBC correlated with lymph node and distant metastasis [89]. Furthermore, the RANTES inhibitors, maraviroc and vicriciroc, reduced invasion and pulmonary metastasis in a preclinical tumor model of breast cancer [90]. IL-6 is a pleotropic cytokine involved in immune regulation and tumorigenesis. One third of total circulating IL-6 originates from adipocytes and circulating levels are correlated with adiposity [91,92]. When adipocytes are cocultured with breast cancer cells, adipocytes increase secretion of IL-6, which in turn promotes invasion and migration of tumor cells [93]. Furthermore, IL-6 also plays a critical role in the biology of CSCs through activation of Notch and JAK/STAT signaling [94,95]. In addition to classical IL-6 signaling, IL-6 trans signaling is a major driver of obesity associated hepatocellular carcinoma (HCC), through inhibition of p53 induced apoptosis and enhanced angiogenesis [96]. IL-6 also promoted HCC progression via upregulation of osteopontin (OPN), a secretory ECM protein involved in the maintenance of the stemness phenotype [97].

The inflammasome is a highly regulated protein complex that triggers caspase-1 activation and subsequent secretion of IL-1β and IL-18 [98]. Obesity related factors, such as cholesterol crystals and fatty acids (palmitate and ceramide), can lead to unchecked activation of the inflammasome [98]. IL-1β expression in primary tumors is a potential biomarker for predicting breast cancer patients who are at increased risk of developing bone metastasis [99,100]. Furthermore, in vitro studies indicate that tumor cells expressing high levels of IL-1β specifically home to and colonize the bone [101]. A persistent increase of circulating levels of TNFα occurs in obesity, mainly due to the elevated number of M1 macrophages in obese WAT [86]. TNFα stimulates the secretion of MMPs in epithelial tumors and enhances EMT to promote invasion and migration of tumor cells [85,102,103]. Furthermore, TNFα increases liver metastasis through inducing expression of cell adhesion molecules, such as ICAM-1, E-selectin and VCAM-1, on liver specific endothelial cells, and thus enhancing tumor cell arrest and transendothelial migration [104,105].

Crown like structures (CLSs) are a histological feature of dead or dying adipocytes surrounded by macrophages and they are increased in obesity associated adipose tissue [37]. CLS are associated with free fatty acid (FFA) release, NF-κB activation and the generation of a proinflammatory microenvironment [106]. They are best characterized for their role in the initiation and progression of breast cancer [106] but they are also reported to play a role in endometrial cancer [107], prostate cancer [108] and non-alcoholic steatohepatitis (NASH)—a major risk factor for HCC [109]. In breast adipose tissue CLS are not only associated with inflammation but they also drive aromatase activity resulting in an increased ratio of estrogen:androgen in blood and local tissues [110]. The expression of programmed death-ligand 1 (PD-L1) in adipocytes prevents the antitumor function of cytotoxic CD8^+^ T cells [43]. It is therefore not surprising that treatment with anti PD-1/PD-L1 immunotherapy in patients with a BMI > 25 (i.e., overweight/obese) have improved clinical outcomes compared to normal weight patients [111].

CSCs will only proliferate in specific tumor environments indicating that environmental stimuli are critical to preserve their phenotypic plasticity, to protect them from the immune system and to facilitate metastatic potential [112]. High levels of proinflammatory cytokines from obese adipose tissue can stimulate CSC properties [113]. TNFα enhances the CSC phenotype in numerous cell types and is associated with upregulation of stem cell related genes, chemo resistance and tumorigenesis [114,115,116]. Furthermore, TNFα upregulates TAZ (a Hippo pathway effector) and Slug (an EMT mediator), which increase breast CSCs through both canonical and non-canonical NF-κB signaling [117,118]. Analysis of TNBC patient datasets reveals high tumor expression of the epigenetic reader methyl-CpG-binding domain protein 2 (MBD2), specifically the alternative splicing variant 2 (MBD2_v2) expression and high relapse rate and high BMI [119]. It is postulated that obesity drives high reactive oxygen species (ROS) levels, which subsequently promotes MBD2_v2 expression and an expansion of the CSC fraction [119].

### 3.3. Angiogenesis 

The processes of angiogenesis and adipogenesis are closely linked. When preadipocytes differentiate into adipocytes and become adipose tissue, new blood vessels are also formed [120]. Conversely inhibition of adipocyte differentiation also reduces angiogenesis [121]. In breast cancer, leptin upregulates all components of the IL-1 system (IL-1α, IL-1β, IL-1Ra and IL-1R tI) and both leptin and IL-1 together promote angiogenesis through expression of VEGF/VEGFR [122]. Furthermore, leptin upregulation of VEGF/VEGFR2 was impaired by IL-1 signal blockage [122]. In addition, obesity leads to an increase in tumor infiltrating macrophages with activated NLRC4 inflammasome and increased IL-1β production in breast tumors [123]. This leads to a NLRC4/IL-1β dependent upregulation of adipocyte derived angiopoietin-like 4 and enhanced obesity associated tumor angiogenesis [123]. Anti VEGF therapy has fallen short of expectations, particularly in breast cancer where the FDA revoked approval for bevacizumab because of a lack of overall survival benefit [124]. Anti VEGF therapy resistance is partly driven through expression of proinflammatory and other alternative angiogenic factors, many of which are also increased in obesity [121,125]. Furthermore, breast cancer patients with obesity are less sensitive to anti VEGF treatment and they have increased systemic concentrations of IL-6 and fibroblast growth factor-2 (FGF-2) [126]. The elevated IL-6 was associated with increased IL-6 production from adipocytes and myeloid cells within tumors and IL-6 blockage abrogated obesity induced resistance to anti VEGF therapy at both primary and metastatic sites [126]. Vasculogenic mimicry (VM) is a tumor vascular system that is independent of angiogenesis of endothelial cells, and is associated with both poor survival in multiple tumor types and anti VEGF therapy resistance [127]. Notably, the adipose tissue secretome has been shown to induce melanoma cells to arrange in 3D vessel like structures, characteristic of vasculogenic mimicry [128], supporting a role of adipose tissue in this process.

### 3.4. Metabolic Repogramming

An essential function of adipocytes is energy mobilization and therefore a metabolic interaction between cancer cells and adipocytes is not surprising. The Warburg effect suggests that due to mitochondrial dysfunction, malignant cells prefer to produce adenosine triphosphate (ATP) via glycolysis instead of oxidative phosphorylation, even in the presence of oxygen [129]. In parallel, cancer cells are able to use alternative sources of energy such as amino acids and lactate from the microenvironment. Bone marrow adipocytes promoted the Warburg phenotype by increased expression of glycolytic enzymes, increased lactate production and decreased mitochondrial oxidative phosphorylation in metastatic prostate cells by paracrine signaling [52]. The “reverse Warburg effect” theory proposes that cancer cells induce oxidative stress in the neighboring stromal cells by secreting ROS and triggering aerobic glycolysis and production of high energy metabolites, especially lactate and pyruvate. These metabolites are then transported through the “lactate shuttle” to sustain the anabolic needs of adjacent cancer cells [130,131]. The effect has been mainly described in stromal fibroblast cells, however given that ASCs and CAA are fibroblast like cells, it is likely they are also important contributors. Indeed it has been reported that the “reverse Warburg effect” was induced during the co-culture of adrenocortical carcinoma cells with ASCs [132]. Ketone bodies are another catabolite produced and released by glycolytic adipocytes and they are an ideal substrate for ATP production by driving oxidative mitochondrial metabolism leading to enhanced tumor invasiveness [91,133].

A key characteristic of CAA is their loss of lipid content. The FFAs released by adipocytes after lipolysis are stored in tumor cells as triglycerides in lipid droplets [134]. Tumor cells then release FFAs from lipid droplets though an adipose triglyceride lipase dependent lipolysis (ATGL) pathway [134]. ATGL is upregulated in tumors on contact with adipocytes and it correlates with aggressiveness by stimulating tumor cell invasion [134]. The FFAs also act as structural units for newly synthesized membrane phospholipids and cancer cell membranes become enriched with saturated and/or mono unsaturated fats leading to changes in membrane dynamics [135]. This results in cancer cells that are more resistant to oxidative induced cell death and reduced the uptake of drugs [135]. Advanced metastatic melanomas frequently grow in subcutaneous tissues largely composed of adipocytes [136]. Adipocyte derived lipids are transferred to melanoma through the lipid transporter FATP1 and a small molecule inhibitor of FATPs reduced melanoma growth and invasion [136]. Furthermore in AML, leukemic blasts activate lipolysis in neighboring bone marrow adipocytes leading to the transfer of lipids to the blast through FABP4 [137].

Amino acids such as glutamine, glycerine, serine and proline also have important roles in the asymmetric metabolism of amino acids between cancer and stromal cells [43]. Glutamine is a pivotal source of the TCA cycle intermediates and ATP in cancer cells, and the substrate of the antioxidant glutathione [138,139]. Stromal cells within the tumor microenvironment harness carbon and nitrogen from non-canonical sources to synthesize glutamine and it is used by the tumor cells to promote growth and metastasis [140]. Glutamine is downregulated in obesity and is inversely associated with proinflammatory gene expression and macrophage infiltration [141]. Pancreatic ductal adenocarcinoma (PDAC) cells rely on glutamine utilization to fulfill their metabolic requirements and it is the most depleted amino acid within the PDAC microenvironment [142]. Glutamine deficiency leads to the induction of EMT through the upregulation of the master EMT regulator Slug [142]. In addition, as glutamine levels decline, tumor cells become more reliant on asparagine for proliferation and protein synthesis [143]. Asparagine affects the metastatic cascade at multiple stages. At the primary tumor level, asparagine promotes EMT and intravasation [144,145]. In the circulation asparagine helps circulating tumor cells survive shear and oxidative stress whilst at a distinct metastatic sites, asparagine facilitates cell growth and survival by inducing glutamine synthetase (GLUL) expression and glutamine biosynthesis [144,145].

Citrulline and nitric oxide are generated by tumor cells by catabolizing the semi essential amino acid arginine [146]. Nitric oxide facilitates glycolytic activity and suppresses oxidative phosphorylation to promote proliferation [146]. Citrulline is secreted into the ECM and is taken up by stromal adipocytes, before being converted back into arginine and released for cancer cells [146]. Depriving tumor cells of arginine has cytotoxic effects through apoptosis or autophagy depending upon the tumor type, and decreasing the ability of tumor cells to migrate and adhere to the ECM [147]. Arginine dependent migration requires arginine to be metabolized by two major enzymes, arginase (ARG1) and nitric oxide synthase (NOS) [147]. In HCC, higher expression of ARG1 is positively correlated to aggressive tumor growth and poor disease free survival [148]. In vitro studies revealed that overexpression of ARG1 enhanced arginase activity leading to multiple processes that contribute to progression including increased cell viability, migration, invasion and EMT [148]. Obesity coupled with PDAC results in accelerated tumor growth and enrichment in pathways regulating nitrogen metabolism. The mitochondrial form of arginase (ARG2) that hydrolyzes arginine into ornithine and urea is induced upon obesity and is accompanied by PDAC growth and increased nitrogen flux from 15N-glutamaine into the urea cycle, the principle pathway for ammonia detoxification [149]. Infusion of 15N-arginine in murine models demonstrates a shunting of arginine catabolism away from the urea cycle into creatine synthesis, resulting in ammonia accumulation specifically in obese tumors [149]. The biological consequences of ammonia accumulation in the tumor microenvironment is not fully understood but it has been shown to directly generate amino acids through glutamate dehydrogenase activity [150]. 

Hyperinsulinemia is a hallmark of chronic obesity and insulin stimulates the hepatic synthesis of the peptide IGF-1 [151]. Three quarters of breast cancer patients show activation of insulin/IGF-1 signaling and this rises to 87% in patients with invasive breast cancer [152]. In preclinical models, blocking IGF in combination with paclitaxel significantly reduced tumor cell proliferation and lung metastasis [152]. Due to the frequent dysregulation of the IGF system in cancer, various components of this system became attractive anticancer targets. However, clinical trials using IGF-1 receptor blocking antibodies failed to meet expectations, except in a small number of malignancies [153,154]. More recent developments reveal that dysregulation of the IGF system results in a substantial expansion of the cancer stem like subpopulation by supporting EMT and self-renewal signaling pathways [153]. IGF signaling regulates these pathways in multiple ways though i) stimulation of the transcription factors of the ZEB and the Snail family implicated in the EMT program, ii) interacting with pluripotency transcription factors (e.g., Oct-4, SOX2, Nanog, p53 and HMGA1 proteins) and iii) regulation of development signaling factors (e.g., Wnt/β-catenin, Notch and Shh pathways) classically involved in cell stemness [153]. Intriguingly, in HER2+ breast cancer patients, high IGF-1 in normal weight patients showed a superior recurrence free survival compared to low IGF-1 [155]. In contrast, high IGF-1 in overweight patients was associated with a reduced recurrence free survival [155]. Obese mice have a heightened inflammatory response in the liver and an increased incidence of metastatic colon carcinoma cells to the liver [156]. Moreover, liver inflammation induced by obesity was abrogated in liver specific IGF-I deficient mice leading to a significant reduction of in liver metastasis [156]. Furthermore, IGF-1 promotes neutrophil polarization to a tumor promoting phenotype and the induction of a prometastatic microenvironment in the liver [157]. 

### 3.5. Extracellular Matrix

In addition to adipocyte hypertrophy and dysregulated lipid metabolism, heightened inflammation, hypoxia and abnormal angiogenesis, obesity is also associated with ECM remodeling. Once a primary tumor is established, cells migrate and invade to form satellite tumors within centimeters of the original tumor mass and/or disseminate through the lymph nodes and vasculature to form secondary macrometastases at distance sites (Figure 1). This progression sequence is dependent on the ability of cancer cells to traverse the ECM. The adipose tissue ECM is a three-dimensional, non-cellular structural support of the numerous cell types that reside in the adipose tissue. It is composed primarily of collagens, fibronectin and to a lesser extent lamins that are supplied by a number of resident cell types including fibroblasts, adipocytes and preadipocytes [158]. In obesity, the highly dynamic adipose tissue ECM is constantly undergoing remodeling and reorganization to accommodate increased adipocyte numbers and adipocyte hypertrophy. A rapid increase in adipose tissue volume can result in regional hypoxia, which triggers excess deposition of fibrillar collagens by adipocytes and myofibroblasts, immune cell infiltration, adipose tissue fibrosis, a desmoplastic stroma and increased tissue stiffness, with overall behavior described as similar to “a wound that never heals”. This state of chronic low-grade inflammation within the adipose tissue drives obesity-associated pathologies including diabetes [159,160], cardiovascular disease [161] and cancer. Indeed, it is very similar to the microenvironment of a thriving tumor mass and thus trophic cancer cells that home to the adipose tissue are well supported by these suitable surroundings. Fibrosis is a hallmark of cancer, and desmoplasia within the tumor microenvironment, is a marker of poor prognosis in cancer [162,163] and can negatively impact drug delivery [164,165]. In the case of PDAC, obesity is associated with aggressive tumors with poor prognosis, and adipocyte accumulation in the malignant pancreas [166]. Obesity-induced accumulation of high adipocyte numbers in the pancreas has been shown to induce inflammation and excessive accumulation of ECM components, i.e., desmoplasia, which promotes tumor progression and resistance to chemotherapy [164].

Adipose tissue is the main component of the breast cancer microenvironment, crosstalk between the breast cancer cells and adipocytes or other adipose stromal cells stimulates the secretion of even larger quantities of ECM proteins, increasing matrix stiffness and scar formation, further enhancing EMT and local invasion of tumor cells. Within the adipose tissue, invading breast cancer cells manipulate adipocytes to form fibroblastic CAAs that secrete large volumes of ECM proteins including collagen I, III and IV, and the cleavage product of collagen IV, endotrophin, which is associated with breast cancer metastatic spread [85,167,168]. Furthermore adipocyte collagen IV has been shown to play a role in the early stages of tumor growth in breast cancer [169]. Recently, it has been shown that breast cancer secreted PAI-1 can stimulate CAA collagen biogenesis and reorganization via the induction of a lysyl hydroxylase protein, PLOD2, facilitating the migration of breast cancer cells along aligned collagen fibers, in vitro and in vivo, further promoting metastasis [170]. Additionally, tumors growing in the adipose tissue-rich microenvironment can induce morphological and functional changes in ADSCs so that they differentiate into a CAF-like myofibroblastic phenotype. Breast cancer cells can induce the differentiation of ADSCs to CAFs involving a mechanism dependent on TGF-β, and these ASC-derived CAF-like cells can promote breast cancer cell motility and invasion in vitro, and expressing high levels of stromal-derived factor 1 (SDF-1), a chemokine associated with a more aggressive and invasive cancer phenotype [171,172]. During obesity-induced fibrosis, as adipocytes become encased in the rigid ECM, necrosis ensues and dysfunctional adipocytes stimulate the recruitment of macrophages to the site, histologically forming CLS around the dying adipocytes [173]. The contribution of these activated macrophages to ECM production most likely occurs through their effects on other cell types. They are a major source of TGF-β, PDGF and other chemokines that attract and activate more ECM protein producing fibroblast type cells to the adipose tissue [174,175]. Finally, fibrosis dynamics are tightly regulated by the metalloproteinase (MMP) protein family, which cleave collagenous fibers, enabling matrix remodeling. There are many MMPs associated with obesity [176], in particular MMP-11 (also known as stromelysin-3/ST-3) has been shown to be overexpressed by adipocytes as a result of stimulation by invading breast cancer cells [177]. MMP-11 is important for collagen VI folding and it is also known to negatively regulate adipogenesis and can dedifferentiate adipocytes so that they can acquire a more fibroblast-like phenotype that benefits the invading tumor cells via involvement in adipose tissue fibrosis and ECM remodeling [45,177].

### 3.6. Extracellular Vesicles

In addition to the milieu of adipose tissue secreted factors discussed in the previous sections, there is now evidence highlighting the role that adipose tissue derived extracellular vesicles (EVs) play in guiding and enhancing the metastatic process. EVs are lipid membrane enclosed particles, measuring on the nanometer scale, that can be classified as either minute exosomes (<100 nM) or larger microvesicles (<1000 nM), which facilitate the horizontal transfer of cellular cargo, including nucleic acid, proteins, lipids and metabolites between communicating cells [178,179]. It is not clear whether obesity alters the content of EVs produced by adipocytes, however in the obese setting larger quantities of adipocyte-derived EVs are secreted compared to lean conditions [180]. While the role of cancer cell-derived EVs in manipulating cells in the tumor microenvironment including adipocytes is well established [181,182,183], very little attention has been given to the bidirectional role EVs play in adipocyte-cancer cell communication, and particularly the influence of adipocyte-derived EVs on tumor cell behavior. Nevertheless, a small number of recent studies demonstrate a link between adipocyte EVs and tumor progression in obesity-driven lung cancer [184], breast cancer [185,186] and melanoma [180,187]. 

One way in which adipocytes can promote tumor progression is through metabolic cooperation, by providing a local supply of fatty acids for the process of fatty acid oxidation (FAO) within tumor cells, an emerging favorable metabolic pathway that enhances tumor invasiveness, proliferation and stem cell properties [180,188,189]. In the later stages of tumor progression once the tumor cells have invaded the adipose tissue, secretion of tumor-derived soluble factors can stimulate adipocyte lipolysis and extracellular release of FFAs into the surrounding microenvironment [39,134,188]. Aside from direct release of FAs from adipocytes, EVs released by adipocytes can also be used as a method to transfer molecules including FA substrates and the protein machinery required for FAO to cancer cells either locally or over larger distances, e.g., through the circulation and other tissues.

Epidemiological studies have identified a link between melanoma aggressiveness and obesity [190,191]. Aptly, subcutaneous adipocytes are one of the main components of the tumor microenvironment of invading melanomas and indeed melanoma cells have been demonstrated to internalize naïve adipocyte (i.e., not previously exposed to cancer cells)-derived EVs, resulting in amplified FAO and an enhanced migratory and invasive tumor cell phenotype [180]. Lazar et al. demonstrated that melanoma cells cultured with adipocyte secreted EVs had an increased ability to form lung metastases in mice xenograft models, with a concomitant upregulation of tumor cell FAO. As previously mentioned obese-derived adipocytes secrete higher numbers of EVs compared to their lean counterparts, thus when comparing the effect of adipocyte EVs derived from obese versus lean conditions, equal concentrations of EV preparations were applied to melanoma cells. Noticeably, in obese conditions only adipocyte-derived EVs elicit a heightened effect on melanoma migration, in addition to enhancing clonogenicity and metastatic potential. Thus differences in the cargo content of these EVs as opposed to the sheer number of vesicles are most likely responsible for this heightened effect. Recent in-depth labeling experiments indicate that around 30% of the proteins within the adipocyte EVs are sufficiently transferred to melanoma tumor cells, and these include proteins involved in EV transport, the transport and storage of FAs, mitochondrial FAO and oxidative phosphorylation [187]. Efforts to understand how an obese state heightens the effect of adipocyte EVs on FAO-induced functions of melanoma cells have revealed that the level of FAO enzymes in melanoma tumor cells is unaltered following uptake of obese versus lean derived EVs. In contrast adipocyte EV supply of FAs and subsequent trafficking to melanoma cells was increased under obese conditions, resulting in enhanced substrate availability and FAO, and the altered mitochondrial dynamics that is critical to melanoma cell migration and invasion. Thus in the obese setting, it is the increased transfer of substrate (e.g., FA) and not machinery (FAO-related enzymes) that enhance FAO in recipient melanoma cancer cells [187].

In whole adipose tissue, aside from mature adipocytes, EVs are found in the supernatant ASCs. While adipocyte EVs are linked to enhanced tumor invasiveness and metastatic potential via lipid metabolism, current literature suggest ADSC secreted EVs play a role in angiogenesis [192,193,194], immune modulation [195,196,197] and tumor development [198]. When exosomes secreted from the preadipocyte cell line, 3T3-L1, are injected into the mammary fat together with breast cancer cells, primary tumor initiation and growth is enhanced. Of particular interest in this review, ASC derived exosomes have been shown to promote proliferation and migration, at least in part through the modulation of Wnt/β-catenin signaling, a key pathway in tumor stemness, EMT and metastasis [199,200]. ASCs are abundant in the microenvironment of highly metastatic osteosarcoma. Recently ASC exosomes has been shown to increase proliferation, invasion and migration of osteosarcoma cells in vitro, and growth and metastasis in vivo, via an induction of COLGALT2, a prometastatic gene that subsequently activates downstream EMT targets vimentin and MMP-2 and -9 [201]. Studies regarding adipose tissue secreted exosomes, be it adipocytes or ASCs, are limited to a small number of studies and a small number of tumor types. Further research is required to understand the role, if any, of adipose tissue-derived EVs in other cancer types, which share an intimate relationship with adipose tissue and are driven by obesity.

## 4. Targeting Metastasis through Adipose Tissue-Tumor Interactions 

Obesity-associated cancer is a substantial public health problem worldwide and correcting the disharmony within dysfunctional adipose tissue represents an opportunity to halt disease progression, as summarized in Figure 2. Due to the importance of adiponectin in the progression of several cancer types, adiponectin receptors agonists have been developed [54]. An adiponectin-based short peptide named ADP 355 has high affinity with AdipoR1 and regulates canonical adiponectin signaling pathways (i.e., AMPK, Akt, STAT3, ERK1/2) to halt tumor growth [202,203]. Adiponectin has a larger role in early stage metastatic processes and there are concerns that its agonists may blunt antitumor immunity. Therefore cancer stage is an important consideration for adiponectin-based agonist therapy [54]. Interestingly, adiponectin may also be modulated with dietary and lifestyle factors. Daily fish intake, omega 3 and fiber supplementation have been shown to increase blood levels of adiponectin [204]. Furthermore, moderate intensity aerobic exercise was shown to elevate adiponectin levels to 260% within one week [205].

Inhibitors of leptin, another important adipokine, are also in development. Conditioned media (CM) from breast adipocytes significantly increased mammosphere formation, a marker of breast CSCs, and depletion of leptin from the CM completely abrogated this effect [206]. In addition, blocking leptin signaling using a full-length leptin receptor (ObR) antagonist also reduced mammosphere formation indicating a potential therapeutic target to block stromal-tumor interactions driving breast CSC-mediated disease progression [206]. The ObR antagonist, Allo-aca, a short leptin-based peptidomimetic inhibits leptin induced tumor cell proliferation and angiogenesis [207,208]. In addition, it has good biodistribution to the brain and could potentially target brain metastasis [209]. 

Adipose tissue significantly contributes to inflammatory cytokines in the tumor microenvironment that may be targeted therapeutically to treat metastasis. Agents targeting IL-6 or the IL-6 receptor have already received FDA approval for the treatment of inflammatory conditions or myeloproliferative neoplasms and are being evaluated in solid tumors [210]. However, despite promising preclinical activity, targeting IL-6 alone in unselected cancer patient populations has not shown benefit to patient outcomes [210]. Therefore, more selective formulations and selecting potentially sensitive patient populations, such as obese patients, are required. A novel-CD44 antibody mediated liposomal nanoparticle loaded with anti-IL6R antibody demonstrated specific antimetastatic efficacy in a preclinical mouse model by inhibiting IL-6/STAT3 signaling in CSCs and also reducing the SOX2^+^ and CD206^+^ cells in the tumor microenvironment of lung metastasis, thus demonstrating dual inhibitory activity on CSCs themselves and the metastatic niche microenvironment [211]. 

Inhibiting IL-1R signaling using the receptor antagonist Anakinra resulted in decreased bone metastasis in an in vivo TNBC model. Pretreatment of Anakinra did not prevent tumor cell numbers that arrived to the bone, but instead held the tumor cells in a dormant state, thereby preventing metastasis [212]. Furthermore, therapeutic treatment of bone metastasis in the model with Ankinira stalled tumor growth by reducing proliferation and angiogenesis [212]. Anakinra and other IL-1 signaling pathway inhibitors are currently in multiple clinical trials to target metastasis in solid tumors, multiple myeloma and plasma cell neoplasm, as summarized in [100]. In a clinical trial of patients with smoldering or indolent multiple myeloma at risk of progression to active myeloma, Anakinra treatment decreased myeloma proliferative rate and hs-CRP in responders leading to improved progression free survival [213]. A large cardiovascular randomized control trial of Canakinumab (Canakinumab Anti-inflammatory Thrombosis Outcomes Study (CANTOS)), an interleukin (IL)-1β inhibitory antibody, noted a dramatic reduction in the incidence of lung cancer [214]. The trial of 10,500 patients was not designed to study lung cancer as an endpoint and the patients were extensively screened prior to recruitment [215]. These results indicate that IL-1β reduced the progression, invasiveness and metastatic spread of early stage lung cancers that were undiagnosed at the time of recruitment [215]. Therefore these studies suggest that anti IL-1β therapy may prevent the metastatic cascade if given early in the disease process, and further studies are required to examine if metastatic spread is also prevented in people living with obesity. However, there is emerging evidence that anti IL-1β therapy may promote metastatic spread if given later in disease development. Castano et al. eloquently reported that primary tumors elicit a systemic inflammatory response involving IL-1β expressing innate immune cells that infiltrate distant metastasis microenvironments [216]. At the metastatic site, IL-1β maintains metastasis initiating cells in a ZEB-1 positive differentiation state, but inhibition of IL-1 receptor relieves the differentiation block and resulted in metastatic colonization [216].

People living with obesity experience more chemoresistance due to altered drug pharmacokinetics and enhanced proinflammatory and adipokine secretions [217]. Therefore further research is required to investigate if inhibitors of downstream intracellular pathways associated with both obesity and chemo-resistance, such as AKT or NF-κB pathways, are clinically superior in combination with chemotherapy in people with obesity. On the other hand, immune checkpoint inhibitors appear to be more effective in the obese population compared to lean patients. The chronic inflammation and lipid accumulation induced by obesity leads to an “exhausted” T cell phenotype, which is associated with PD-1 upregulation [218]. Obese melanoma patients treated with immune checkpoints inhibitors have an improved overall survival and progression free survival compared to lean patients [219]. In addition, response to PD-1 inhibitors has been modest in TNBC however results from the JAVELIN trial suggests that obese TNBC patients with an exhausted immune response might benefit from checkpoint blockage [220,221,222]. However, aged mice with excess adiposity developed a lethal cytokine storm reaction following immunotherapy, potentially indicating that older people with obesity may be at a higher risk of adverse effects [223].

Adipose derived fatty acid binding proteins (FABPs) in the tumor microenvironment have important roles as intracellular lipid chaperones and pharmacological agents that modify FABP function are in development for obesity, diabetes and atherosclerosis [224]. FABP5 is not expressed in the normal prostate but is highly upregulated in metastatic prostate cancer and offers is a novel therapeutic target for pharmaceutical inhibition [225,226]. However, lipid metabolism is central to the function of regulatory T cells (Tregs). Inhibition of FABP5 in Tregs triggers mitochondrial DNA release, which induces cGAS-STING dependent type 1 interferon signaling and the production of IL-10. Overall this promotes Tregs immunosuppressant activity within the tumor microenvironment leading to enhanced tumor progression [227]. Treg activity is associated with a poor prognosis and therefore the effect of pharmacological inhibition of FABP5 would have to be carefully evaluated in all cells within the tumor microenvironment [228].

CD36 is a transmembrane glycoprotein that mediates lipid uptake and binds to a diverse range of ligands including apoptotic cells, thrombospondin-1 (TSP-1) and fatty acids [229]. High fat diet or palmitic acid enhances a subpopulation of CD36^+^ cells with metastatic potential in mouse models of human oral cancer [230]. The presence of CD36^+^ metastasis initiating cells also correlated with poor prognosis in a number of cancer types and anti-CD36 neutralizing antibodies induced regression of the lymph node and lung metastasis in murine models [230]. However, there is also evidence that loss of CD36 in stromal cells and surrounding endothelial cells can increase the metastatic potential of the niche and induce angiogenesis [231,232]. Furthermore, several agents targeting CD36 in clinical trials with thrombospondin mimetic peptides have been unsuccessful due to adverse effects and limited efficacy [229,232]. CD36 may be promising to specifically target metastasis however further knowledge of its role in stromal cells and downstream signaling pathways is required to develop more precise therapeutics.

As glutamine levels are decreased in obesity and cells adapt to low glutamine by becoming dependent on asparagine, an amino acid, which promotes multiple stages of the metastatic cascade, targeting asparagine may prevent obesity-induced metastasis [141,143,145]. Indeed, treatment with L-asparaginase, an enzyme that depletes asparagine, or alternatively dietary asparagine restriction, reduced metastasis in a breast cancer model without affecting the growth of the primary tumor [145]. L-asparagine is a mainstay treatment in childhood acute lymphoblastic leukemia (ALL) and children with obesity have a 50% greater risk of ALL relapse than lean counterparts [233]. Glutamine synthesis was markedly increased in bone marrow adipocytes following chemotherapy and obesity substantially impaired L-asparaginase efficacy due to adipocytes secreting glutamine [233]. In addition, obesity increases the risk of hepatic toxicity with L-asparaginase therapy by promoting a maladaptive integrated stress response [234]. Therefore, whilst inhibiting asparagine preferentially targeted metastasis, the experience of L-asparaginase in ALL suggests that it could be less effective and more toxic in people with obesity.

Another proposed method to target the metabolism of tumor cells is through short term dietary fasting. Fasting diets are currently in clinical trial to enhance chemotherapy efficacy and reduce toxicity however there is also evidence they may slow tumor progression [235]. In the fasting state IGF-1, a promoter of EMT, is inhibited, and clinical trials are underway to investigate if tumor progression can be inhibited [235]. Obesity induces lung neutrophilia leading to increased breast cancer lung metastasis through GM-CSF and IL-5 signaling. Interestingly, weight loss reversed this effect, leading to reduced serum GM-CSF and IL-5 in both mouse and humans. In addition, this suggests that the disruption of normal lung homeostasis is critical in obesity induced metastasis and pharmacological interventions that resolve lung inflammation may reduce metastatic risk [236]. Targeting IL-5 was shown to be sufficient to block lung neutrophil trafficking in obese mice and there are a number of anti-IL-5 therapies with FDA approval for severe hypereosinophilic asthma [236,237]. Therefore, obese cancer patients with increased neutrophilia, GM-CSF or IL-5, may benefit from anti-IL-5 therapies to prevent lung metastasis development. 

## 5. Targeting Metastasis via the Treatment of Obesity

Complementary treatment of obesity may potentially improve outcomes in cancer patients [238]. Indeed, a systematic review and meta-analysis of animal models concluded that weight loss decreases cancer progression and metastasis [239]. The most common method of weight management is dietary calorie restriction (CR) and increasing energy expenditure though exercise. CR extends lifespan and reduces age related diseases, including cancer, in preclinical models and recently translational studies have tested the potential of CR as an adjuvant therapy in cancer. However, chronic CR is often contra indicated in cancer patients due to the risks of malnutrition and cachexia and other strategies, such as intermittent fasting, CR mimetic drugs or alterative diets (e.g., ketogenic diet) may be more suitable [240,241]. Clinical studies have shown that both adjuvant and neo-adjuvant chemotherapy is less effective in people with obesity [242,243,244]. CR, fasting or fasting mimicking diets (FMDs) may improve response to both chemotherapy and radiotherapy as well asreducing side effects associated with cytotoxic drugs [240]. In addition, obesity is associated with an increase desmoplasia surrounding the tumor tissue, contributing to chemoresistance, and CR can reduce the desmoplasia to facilitate better therapeutic drug delivery to the tumor cells [164,240]. CR, fasting and FMD also have been shown to reduce secreted factors within the tumor microenvironment including proinflammatory cytokines, leptin and IGF-1 [241]. Considering the key roles of these molecules to be dysregulated in obesity and promote metastasis, CR, fasting or FMD may reduce tumor progression through this mechanism. Interestingly, however, a preclinical study of colon cancer showed that an intermittent fasting diet expanded the CSCs population and enhanced EMT by promoting the Warburg/Crabtree effect following post-fasting food overconsumption [245]. Therefore, carefully controlled clinical studies of CR, fasting and FMD are required to determine if they have a role in preventing cancer progression without leading to nutritional deficiencies or cachexia. With this in mind, weight management strategies may be a more appropriate strategy to prevent reoccurrence during long-term cancer survivorship.

Physical inactivity is a well-known avoidable risk factor for developing cancer. However, there is also evidence that an increased level of physical activity between before to after diagnosis reduces cancer mortality [246]. In contrast, a decreased physical activity level between pre- to post- diagnosis is associated with higher cancer mortality [246]. Together this indicates that tumor progression could be slowed or prevented by physical activity and indeed physical activity regulates multiple steps of the metastatic cascade, as comprehensively reviewed in [247]. In summary, moderate intensity exercise appears to prevent tumor spread by normalizing angiogenesis, reducing circulating tumor cells and decreasing endothelial cell permeability [247,248,249]. On the contrary, high-intensity exercise may favor cancer dissemination and promote the formation of the premetastatic niche through excessive stress [247,250]. In overweight or obese breast cancer survivors, moderate to vigorous exercise three times a week for 16 weeks reduced systemic levels of insulin, IGF-1 and leptin, and increased adiponectin [251]. Therefore, chronic adaptations to moderate-intensity endurance exercise may be the most effective way to achieve a preventive effect of exercise on metastases [247]. Overall, exercise and nutritional strategies may be especially beneficial in overweight or obese cancer survivors to reduce both obesity-associated comorbidities and prevent cancer reoccurrence.

Bariatric surgery is the most effective treatment for obesity and its associated metabolic and cardiovascular complications. On the whole studies report that overall cancer incidence and mortality is reduced post bariatric surgery, particularly in women [252,253,254]. Obesity is responsible for increased estradiol production in adipose tissue thus driving hormone sensitive cancers and the augmented concentration of total serum estrogen following bariatric surgery is suggested to be the reason for reduced post-menopausal breast and endometrial cancer risk [254]. Indeed bariatric surgery may also reduce tumor progression by specifically treating hyperinsulinemia and reducing high circulating IGF-1 levels [254]. However, data is conflicting and there is a potential increase in cancer incidence following bariatric surgery [254,255]. There are reports suggesting bariatric surgery results in an increased risk of colorectal cancer (CRC) but the results are complicated as control groups were not standardized for weight changes, dietary changes and medications, which can have a profound impact on CRC incidence [255]. One hypothesis for the increased CRC incidence is based upon a study investigating colonic epithelial cell proliferation in subjects undergoing roux-en-Y-gastric bypass surgery [256]. Biopsies comparing the day of surgery and 6 months after surgery found an increase in the mitotic activity of colonic cells following surgery and an increased expression of proinflammatory and tumorigenic COX-1 and COX-2 enzymes [256]. Bariatric surgery can successfully achieve weight loss among cancer survivors, but long term effects have not been reported [257]. Overall, current evidence indicates bariatric surgery is more effective for cancer prevention in female hormone driven cancers compared to male cancers. Indeed, a study of women with BMI > 40 kg/m^2^ who underwent bariatric surgery resulted in a reduction of endothelial proliferation marker and resolved most cases of the precancerous condition atypical hyperplasia [258]. Randomized control trials are required to determine if bariatric surgery in obese women can increase breast or endometrial cancer survivorship and reduce tumor reoccurrence.

## 6. Conclusions

Obesity is a growing and significant global health crisis, with epidemiological studies demonstrating a compelling association between cancer risk and progression. A clearer understanding of the molecular processes associated with obesity-induced cancer is warranted and has the potential to uncover new targeting strategies. Indeed, potential new targets are already emerging. It is reasonable to posit that weight loss might be beneficial to reduce metastasis risk. However, it is also possible that weight loss alone is insufficient to reverse the effects of chronic obesity due to potential epigenetic reprogramming of tumor cells and/or stromal cells. In addition, patients with advanced disease often experience WAT loss and wasting (cachexia), which complicates treatment and adversely affects patient survival. The Breast Cancer Weight Loss (BWEL) trial (Identifier NCT02750826) is a phase III randomized trial currently underway assessing the impact of a weight loss intervention on disease recurrence in women with stage II to III HER2-negative breast cancer. The results of this trial will help provide answers to some of these questions. Given the limitations of BMI as a measurement of adiposity, novel methods of identifying people with metabolically unhealthy adipose tissue is required. Additional strategies may also benefit individuals who are not obese (BMI< 30) but who have signs of metabolic syndrome or adipose tissue dysfunction. Ultimately, there is a definitive need for more tailored interventions for obese patients to improve survival status in this patient population, especially considering the higher risk of drug resistance in the obese setting.

## Figures and Tables

**Figure 1 cancers-12-03798-f001:**
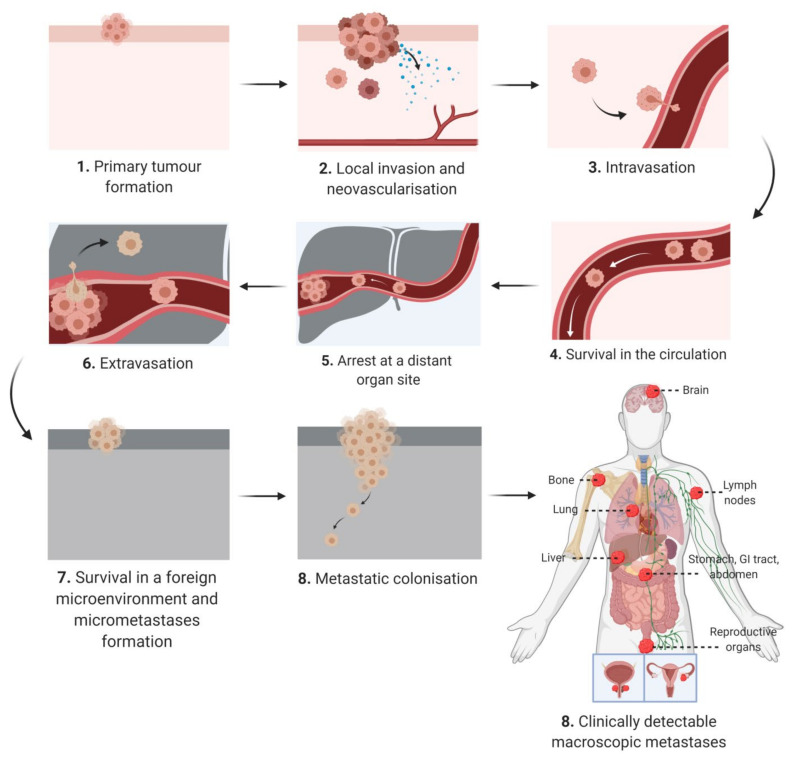
Schematic diagram depicting the keys steps of the metastatic cascade from initial presentation as an in situ tumor mass at the primary site to macroscopically detected metastatic lesions at secondary sites.

**Figure 2 cancers-12-03798-f002:**
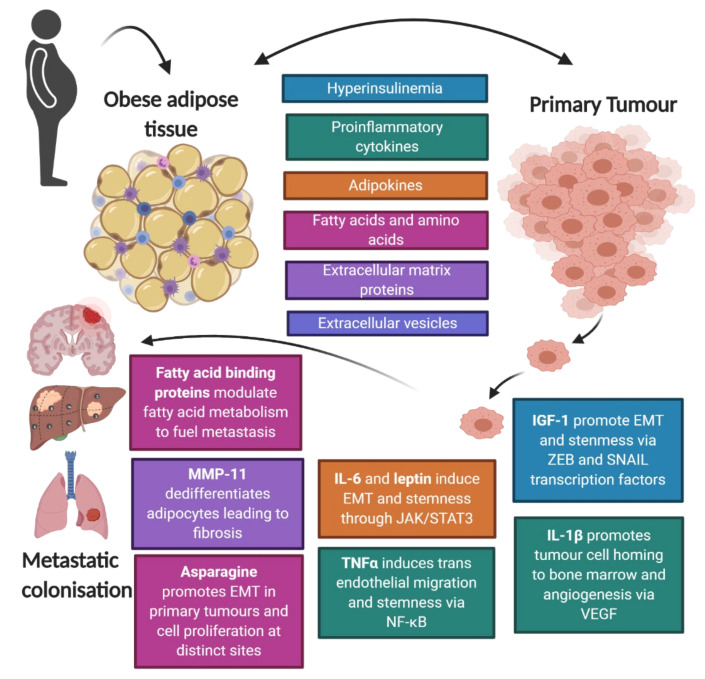
Summary of cross talk between adipose tissue and cancer cells that promote metastasis. Hyperinsulinemia in obesity induces insulin growth factor (IGF)-1 signaling, which promotes epithelial to mesenchymal transition (EMT) and cancer stem cells (CSCs) though ZEB and SNAIL transcription factors leading to an increase in pluripotency transcription factors (OCT4, SOX2 and NANOG) and the developmental signaling pathways (Wnt and NOTCH). Interleukin 1β (IL-1β) promotes promote tumor cell homing to the bone marrow and angiogenesis through vascular endothelial growth factor (VEGF) signaling. Interleukin 6 (IL-6) and leptin induces epithelial to mesenchymal transition (EMT) and CSCsthough JAK/STAT3 signaling. Tumor necrosis factor α (TNFα) induces trans endothelial migration and CSCs though nuclear factor kappa-light-chain-enhancer of activated B cells (NF-κB) signaling. Fatty acid binding proteins (FABP) modulates fatty acid metabolism from neighboring adipocytes to provide fuel for tumor cells to metastasis. Matrix metalloproteinase 11 (MMP-11) dedifferentiates mature adipocytes into cancer associated adipocytes (CAAs) leading to extra cellular matrix (ECM) remodeling and fibrosis. Asparagine promotes EMT in primary tumors and increases tumor cell proliferation at distinct metastatic sites.

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
