# Peer review of "Obesity and Cancer Metastasis: Molecular and Translational Perspectives"

_cancers, 2020, doi:10.3390/cancers12123798_

Round 1
Reviewer 1 Report
This is a good comprehensive review on the link between obesity and cancer metastasis. Several molecular mechanisms are covered in detail and appropriate references provided.
I suggest, however, that the authors include one additional chapter discussing how obesity-targeting strategies (pharmacologic, surgical, and lifestyle modifications) can modulate cancer risk and, especially, metastatic cancer progression in humans. This could also tackle the impact of obesity and overweight on the response to chemotherapy and biologic therapy.
Author Response
Many thanks for your time in reviewing the manuscript and your valuable feedback. We have included a new section based outlining strategies to treat obesity and their potential role in targeting metastasis. Please see revised manuscript.
Reviewer 2 Report
This review manuscript by Annett, Moore, and Robson carefully summarizes the current knowledge of the field. This review is extensive and complete.
This review has very minor comment as below.
Minor comments:
1) There are only a few errors in the reference list. Some of reference information is incomplete.
Author Response
Many thanks for your kinds comments and your time in reviewing out manuscript. We have further proof read the manuscript and made small typological changes. We utilised a reference managing software to compile the reference list, would you mind outlining which references require further changes?
Reviewer 3 Report
Overall, the authors Annett, et al, have produced a timely and detailed review in into the mechanisms and implications between cancer progression and obesity, and highlighted many roles of adipose tissue in promoting a favorable tumor microenvironment. It will be well received.
I would encourage the authors to briefly highlight in addition the inflammatory and tumorgenic nature of autotaxin and it's product lysophophatidic acid, primarily because there is a first in-class inhibitor that is currently in phase III trials for pulmonary fibrosis, and is expected to act as a potent adjuvant treatment for chemo/radiotherapy by breaking the microenvironment communication between adipose tissue and cancer cells. Links to few relevant reviews are below
https://pubmed.ncbi.nlm.nih.gov/31179008/
https://pubmed.ncbi.nlm.nih.gov/32041123/
https://pubmed.ncbi.nlm.nih.gov/31934128/
https://pubmed.ncbi.nlm.nih.gov/30332900/
Author Response
We would like to thank the reviewer for their positive comments and their time for reviewing the manuscript. Thank you for your suggestion to add in a paragraph on the role of autotaxin in cancer metastasis. We have modified the text and believe this has enhanced the overall manuscript.